# Fracture Evolution of Overburden Strata and Determination of Gas Drainage Area Induced by Mining Disturbance

**Yuchu Cai** [1,2], **Shugang Li** [2,3,*], **Xiangguo Kong** [2,3,*], **Xu Wang** [4], **Pengfei Ji** [2,3], **Songrui Yang** [2,3], **Xi Lin** [2,3], **Di He** [2,3] **and Yuxuan Zhou** [2,3]

1   School of Chemistry and Chemical Engineering, Xi'an University of Science and Technology, Xi'an 710054, China
2   College of Safety Science and Engineering, Xi'an University of Science and Technology, Xi'an 710054, China
3   Key Laboratory of Western Mine and Hazard Prevention, Ministry of Education of China, Xi'an 710054, China
4   Shandong Anke Xingye Intelligent Equipment Co., Ltd., Jinan 250002, China
*   Correspondence: lisg@xust.edu.cn (S.L.); kxgtudou7218@xust.edu.cn (X.K.)

**Abstract:** Overburden strata fracture evolution is critical to dynamic disaster prevention and gas-relief drainage, so it is important to accurately determine the evolution relationships with mining disturbance. In this paper, experiments and numerical simulation were adopted jointly to characterize the time-varying fracture area of overlying strata. The experimental results showed that the roof strata gradually broke and collapsed with coal mining, which indicated the fractures of overburden strata developed in an upward direction. The fracture development causes were explained by numerical simulation, which showed that stress increase exceeded the strength of coal and rock strata, and fractures were formed and expanded. Both experiments and numerical simulation results showed the two sides and the top of fracture areas provided channels and spaces for gas migration and reservoir, respectively. In addition, the breaking angle of overburden strata and the height of fracture areas were analyzed quantitatively. Through microseismic monitoring at the mining site, the fracture scales and ranges of overburden strata were verified by the energy and frequency of microseismic events, which were consistent with the support of maximum resistance. The position of drainage boreholes was considered based on the results of overburden strata fracture evolution. Our study is aimed at promoting coal mining in safety and improving gas drainage with a sustainable approach.

**Keywords:** fracture evolution; mining disturbance; experimental analysis; numerical simulation; microseismic response

## 1. Introduction

During coal mining, the overburden strata gradually bend, sink and break [1]. Along the direction of formation height, there are collapse zones, fracture zones and bending subsidence zones formed in overburden strata. Especially in fracture zones, this provides space for gas storage [2–6]. To prevent a gas disaster, it is necessary to conduct gas drainage in this area. The extracted gas can be used as clean energy, which can improve resource utilization efficiency and avoid disasters [7–9]. However, it is a difficult problem to determine the exact location of the overburden fractures caused by mining disturbance.

The processes of fracture initiation, propagation and penetration of coal and rock specimens under different loading conditions have been studied [10–14]. Yang et al. (2020) studied deformation characteristics and fracture evolution of a jointed rock mass, and the relationship between these and joint angles [15]. Li et al. (2022) investigated fracture evolution and failure behavior of granite samples with cross-joints in uniaxial loading experiments, and found that cross-joints affected strength and deformation properties [16]. In cyclic loading experiments, Wang et al. (2022) explored fatigue damage and fracture evolution of red sandstone, and concluded that crack propagation was dominated by tensile cracks [17]. Pirzada et al. (2021) discussed contact area and aperture evolution of

natural fractures during the shearing process of rock [18]. Due to the discrete properties of specimens, a large number of tests are required in studies to determine common processes in the evolution of fractures [19–21].

With respect to discrete properties of samples, numerical simulations have been used in the study of coal and rock fracture evolution [22,23]. Han et al. (2020) divided the shearing process of rock-like materials into four stages, including the linear elastic stage, crack strengthening stage, plastic softening stage, and residual strength stage, based on the simulation results [24]. Ju et al. (2019) analyzed stress field evolution during the fracture development process using a numerical solution with discrete elements, which indicated mining effects could be predicted [25]. A large deformation in the mining space was discovered. Vazaios et al. (2019) found that pre-existing joints in the rock mass govern the fracturing mechanism [26]. Further, Park et al. (2022) explained that stress redistribution was the cause of damage evolution and fracture formation [27]. Based on rock slope engineering, Bouissou et al. (2012) investigated fracture evolution and deformation characteristics [28].

To reveal overburden fracture evolution, Yin et al. (2016) built a multi-field coupling coal mine dynamic disaster simulation test system, from which the morphology of fractures in different overburden layers was obtained [29]. Due to different mining conditions, the processes of fracture evolution in overlying rock strata are different. Qiao (2017) researched crack evolution of overlying strata during fully mechanized top-coal caving at a site with a high mining height, and built an empirical formula for the development height of fracture zones [30]. Zhao et al. (2021) studied fracture evolution of overlying strata in a shallow-buried underground mining site with an ultra-high working face (8.8 m), and investigated crack formation mechanisms based on energy dissipation theory [31]. During multi-seam mining conditions, a coal seam with low risk is preferred as the first protective layer for mining [32]. Jiao et al. (2017) analyzed overburden strata movement and fissure evolution in a lower protective mining layer, and found that mining disturbance would affect fracturing of the overburden coal seam [33]. Therefore, fractures of the overlying strata, resulting from repeated mining, would become more developed. With respect to repeated mining of coal seams that are close together, Yang et al. (2022) used fractal dimensions, fracture entropy and fracture rate to quantitatively reflect spatial and temporal characteristics of overburden fractures [34]. With the improvement of support equipment, the mining of extra thick coal seams is taking place, especially in Shaanxi Province, China. Li et al. (2022) summarized the relationship between overburden fractures, mining heights, advancing speeds and dip angles, and established a mathematical model of a compacted area in an elliptical belt affected by the above factors [35]. However, how to determine overburden fracture evolution under these conditions it is not completely clear.

In terms of overburden fracture evolution in an extra-thick coal seam, experiments and numerical simulations were used jointly to characterize the fracture development area. By experiments, the spatial distribution processes of overburden fractures were analyzed across different mining distances, then stress changes were investigated to explain the causes of fracture evolution by numerical simulation. Concerning fracture evolution analysis, Xiao et al. (2022) studied the microseismic responses during overburden fracture evolution, and dense cracks areas were monitored by the energy and frequency of microseismic events [36]. Microseismic monitoring methods were also used to analyze fracture evolution around the mining space in this study. At the mine site, spatial and temporal measurements of microseismic events confirmed the results of experimental and numerical simulations. Our research results will improve determination accuracy of fracture areas, which will provide the basis for gas extraction borehole layout.

## 2. Engineering Background

In the Shaanxi Binchang Xiaozhuang Mining Co., Ltd., Binzhou city, Shaanxi, China, the main coal seam is 0.8~35.22 m thick, and the average thickness reaches 18.01 m. During coal mining large amount of gas may be emitted. Gas flow into the mining space leads to

possible serious gas explosions and other accidents. Therefore, it is necessary to drain gases during coal mining. To improve gas drainage efficiency, it is important to determine the position of extraction boreholes that are closely related to cracks area of overburden stratum.

In terms of this engineering problem, the evolution of cracks in the overburden of an extra-thick coal seam was studied, and the test location was selected at the working face (No. 40205) in Shaanxi Binchang Xiaozhuang Mining Co., Ltd. The strike and dip length of the test working face are 2007 m and 196 m, respectively. This is adjacent to the mined area (No. 40204 working face) at south side, while the north side is the working face of No. 40207, is undisturbed. During coal mining at this site, a fully mechanized top coal caving method was adopted, and the roof was managed by the full caving method. The floor elevation of the coal seam is +370 m~ +386 m. In geological engineering, the strength of the coal and rock strata, and the gas within them, influence mechanical properties, related to the lithology and thickness of overburden strata [37,38]. The lithology and thickness of the upper and lower strata of the coal seam are shown in Table 1, and are decisive factors of overburden fracture evolution.

**Table 1.** Rock stratum distribution and their ratios of geology and experimental model.

| No. | Lithology | Actual Thickness/m | Model Thickness/cm | Ratio | Sand /1 cm | Gypsum /1 cm | Large White Powder/1 cm | Coal Ash/1 cm |
|---|---|---|---|---|---|---|---|---|
| 33 | Conglomerate | 52.13 | 26.0 | 955 | 8.66 | 0.48 | 0.48 | |
| 32 | Sandy mudstone | 14.80 | 7.5 | 955 | 8.65 | 0.48 | 0.48 | |
| 31 | Sandy mudstone | 20.20 | 10.0 | 955 | 8.65 | 0.48 | 0.48 | |
| 30 | Sandy mudstone | 33.59 | 16.5 | 955 | 8.65 | 0.48 | 0.48 | |
| 29 | Sandy mudstone | 3.6 | 2.0 | 955 | 8.65 | 0.48 | 0.48 | |
| 28 | Coarse gravelly sandstone | 3.80 | 2.0 | 828 | 7.69 | 0.19 | 0.77 | |
| 27 | Siltstone | 3.95 | 2.0 | 837 | 7.69 | 0.29 | 0.67 | |
| 26 | Coarse gravelly sandstone | 0.8 | 0.5 | 828 | 7.69 | 0.19 | 0.77 | |
| 25 | Siltstone | 3.95 | 1.5 | 837 | 7.69 | 0.29 | 0.67 | |
| 24 | Siltstone | 5.80 | 3.0 | 837 | 7.69 | 0.29 | 0.67 | |
| 23 | Mudstone | 3.00 | 1.5 | 828 | 7.69 | 0.19 | 0.77 | |
| 22 | Fine grained sandstone | 2.10 | 1.0 | 828 | 7.69 | 0.19 | 0.77 | |
| 21 | Coarse grained sandstone | 5.72 | 2.5 | 828 | 7.69 | 0.19 | 0.77 | |
| 20 | Sandy mudstone | 9.40 | 4.5 | 955 | 8.65 | 0.48 | 0.48 | |
| 19 | Mudstone | 1.84 | 1.0 | 828 | 7.69 | 0.19 | 0.77 | |
| 18 | 1# Coal | 3.24 | 1.5 | 946 | 4.33 | 0.38 | 0.58 | 4.33 |
| 17 | Mudstone | 1.84 | 1.0 | 828 | 7.69 | 0.19 | 0.77 | |
| 16 | Sandy mudstone | 9.10 | 4.5 | 955 | 8.65 | 0.48 | 0.48 | |
| 15 | Sandy mudstone | 10.55 | 5.0 | 955 | 8.65 | 0.48 | 0.48 | |
| 14 | 3# Coal | 1.85 | 1.0 | 946 | 4.33 | 0.38 | 0.58 | 4.33 |
| 13 | Sandy mudstone | 10.55 | 5.0 | 955 | 8.65 | 0.48 | 0.48 | |
| 12 | Sandy mudstone | 6.15 | 3.0 | 828 | 7.69 | 0.19 | 0.77 | |
| 11 | Siltstone | 3.20 | 1.5 | 837 | 7.69 | 0.29 | 0.67 | |
| 10 | Fine grained sandstone | 7.00 | 3.5 | 828 | 7.69 | 0.19 | 0.77 | |
| 9 | Mudstone | 1.00 | 0.5 | 828 | 7.69 | 0.19 | 0.77 | |
| 8 | 4–1# Coal | 1.15 | 0.5 | 946 | 4.33 | 0.38 | 0.58 | 4.33 |
| 7 | Mudstone | 1.00 | 0.5 | 828 | 7.69 | 0.19 | 0.77 | |
| 6 | Fine grained sandstone | 3.97 | 2.0 | 828 | 7.69 | 0.19 | 0.77 | |
| 5 | Sandy mudstone | 4.00 | 2.0 | 955 | 8.65 | 0.48 | 0.48 | |
| 4 | Coarse gravelly sandstone | 2.80 | 1.5 | 82 | 7.69 | 0.19 | 0.77 | |
| 3 | Fine grained sandstone | 2.48 | 1.5 | 828 | 7.69 | 0.19 | 0.77 | |
| 2 | Sandy mudstone | 2.87 | 1.5 | 955 | 8.65 | 0.48 | 0.48 | |
| 1 | 4# Coal | 15.00 | 7.5 | 946 | 4.33 | 0.38 | 0.58 | 4.33 |

## 3. Experiments

### 3.1. Experimental Parameters

A test platform (3000 × 200 × 1250 mm) was selected for the coal mining experiments. Based on similarity theory, similar conditions need to be determined by comparison with actual geological conditions in the mine (Table 1). The constants in our experiments are shown in Table 2, and the materials (sand, gypsum, large white powder and coal ash) proportions of the rock strata and coal seams in the experimental model as shown in Table 1.

**Table 2.** Similarity constants of the model.

| Along Coal Seam Direction | Model Size mm × mm × mm | Similarity Constant | | | | | |
|---|---|---|---|---|---|---|---|
| | | GeomeTry/$\alpha_L$ | Time /$\alpha_t$ | Bulk Density /$A_y$ | Poisson's Ratio /$A_u$ | Stress /$\alpha_\sigma$ | Strength/$\alpha_E$ |
| Strike | 3000 × 200 × 1250 | 200 | 14.14 | 1.5 | 1.0 | 300 | 300 |

### 3.2. Experimental Model

The similar experimental materials used in the experiments mainly included sand, gypsum, white powder, and coal ash. When producing the model, mica was evenly spread on each layer as a weak surface layer. The steps for constructing the model were as follows: (1) According to the calculated amount of each layered material in the experimental model, the corresponding ingredients were weighed and loaded in a mixing device. (2) The dry materials were mixed evenly, and then an appropriate amount of water was poured into the prepared materials and mixed immediately to prevent agglomeration. (3) Next, the evenly mixed material was poured into the model support, and tamped with iron blocks to maintain the required bulk density. (4) A wall knife was used to mark natural cracks on the surface of the simulated rock stratum at an interval of about 10–20 mm, and a layer of mica powder was evenly sprinkled on the surface to simulate the layer level, and then smoothed by the wall knife again. (5) The other rock layers were paved in sequence in a similar way until all rock layers were prepared on the test platform. (6) The model was erected and left standing for 2–3 days, then U-shaped steel on the surface of model was removed and the model was placed in a ventilated and dry place. After about 20–30 days, the preliminary treatment of model surface was carried out before coal seam mining experiments. (7) When the thickness of rock stratum on the top boundary could not be simulated, a counterweight was added.

### 3.3. Experimental Procedure and Results

To eliminate the boundary influence on coal seam excavation, coal pillars (20 cm) were reserved at the left and right sides of the model. Near the left coal pillar, the open-off cut (4.5 cm) was dug and mined at intervals of 3 cm. When the mining was advanced to 65 m, the self-weight of rock stratum and the uncollapsed part of the roof formed a cantilever beam that transmitted pressure on the basic roof, which would otherwise break and collapse under the initial weigh [35,36]. A collapsed morphology is shown in Figure 1a, from which it can be seen that the caving height is 12.9 m, and the left and right breaking angle reach 70.9°and 64.4°, respectively. After coal mining advanced to 95 m, the first periodic weighting occurred on the overlying strata with a weighting step distance of 30 m. From Figure 1b, it can be seen that the caving height was 30 m from the coal seam floor, and the cavity height was 13.1 m, which would act as gas storage area. In addition, the left and right breaking angle decreased to 52.4°and 51.9°, respectively. When the working face was advanced to 131 m, the second cycle of weighting occurred with a weighting step distance of 36 m. The caving height was 53.7 m away from coal seam floor, and the parting fractures tended to develop in an upward direction. In Figure 1c, the breaking angles of the working face side and the cut side were 54.1°and 54.3°, respectively. There were still large parting fractures above the middle of goaf, and the temporary gas storage area reached 10.1 m high. At a mining distance of 161 m, the overall movement of overlying strata produced the third periodic weighting, as shown in Figure 1d. The height of the crack zone reached 64.3 m, when, and the height of caving zone was 37.1 m. Overburden movement caused the central goaf to collapse and become compacted, while the fractures at the cut-off side and the working face side were highly developed, to become the dominant channel of gas migration. With coal mining, the collapse of the overburden rock and the formation of the fracture zone changed regularly and periodically. For example, the fourth periodic weighting occurred at a mining distance of 191 m (Figure 1e) and the fifth cycle

of weighting occurred at a mining distance of 239 m (Figure 1f). Fracture evolution, gas migration channels, and storage space, gradually changed with time [2,3].

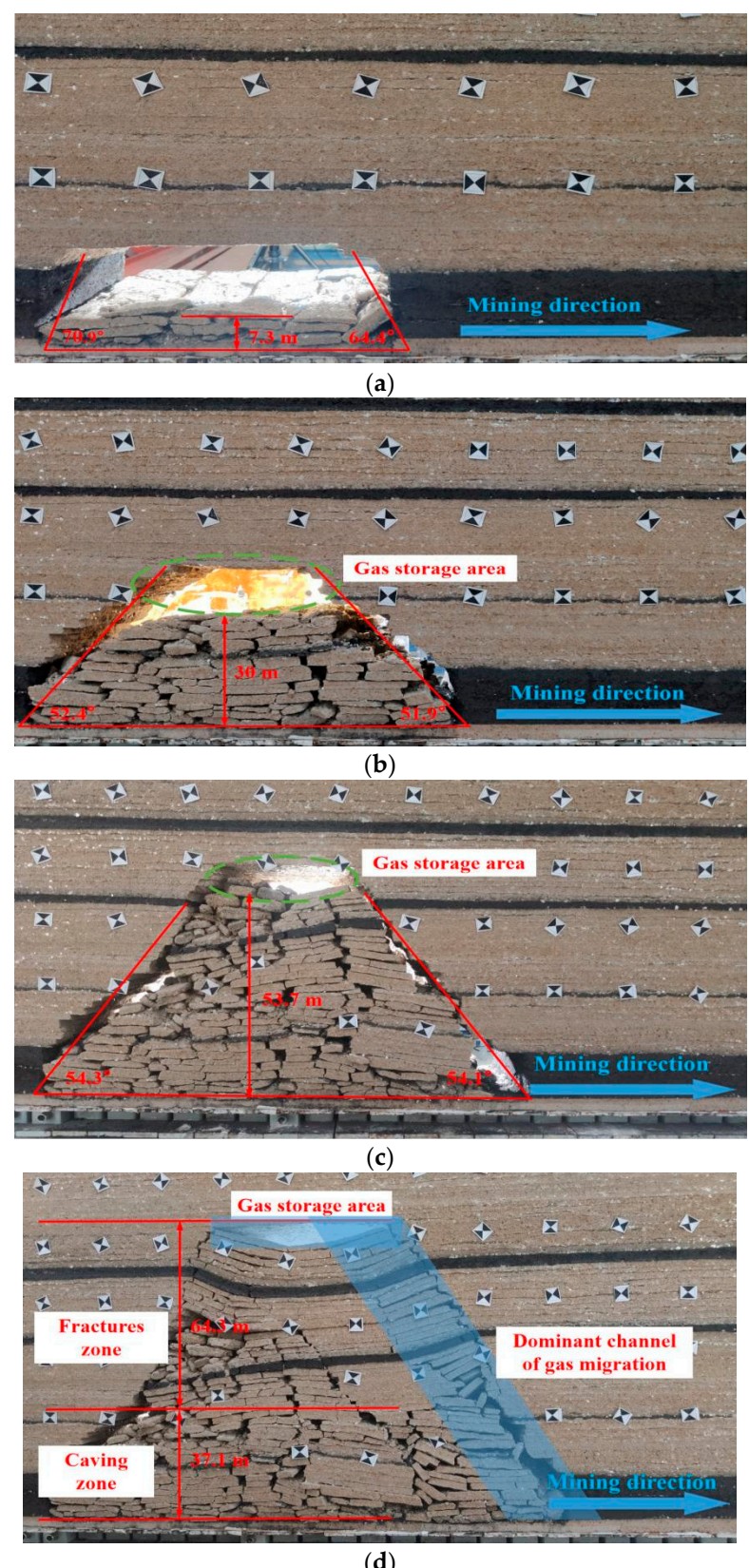

**Figure 1.** *Cont.*

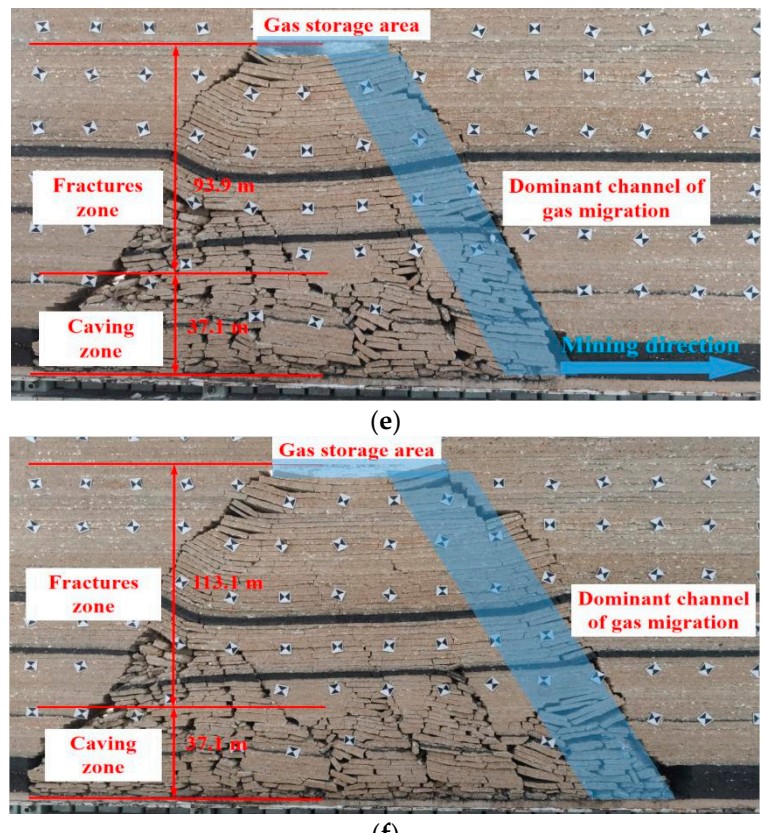

**Figure 1.** Fracture evolution, gas migration and storage area formation of overburden strata in experiments. (**a**) Mining distance of 65 m, (**b**) mining distance of 95 m, (**c**) mining distance of 131, (**d**) mining distance of 161 m, (**e**) mining distance of 191 m, (**f**) mining distance of 239 m.

## 4. Numerical Simulation

Experimental results can provide basic data for studying fracture evolution in the mining overburden, but a large-scale prototype simulation cannot be achieved. To further investigate the evolution of coal mining fractures from large-scale tests, discrete element numerical simulation software was used to study fracture processes [39–42]. In our study, 3DEC was used to establish a numerical model to study spatial fracture distribution and stress characteristics during coal mining.

### 4.1. Geometric Model, Boundary Conditions and Parameter Setting

Based on the actual geology condition of the coal seam, a geometric model (300 m × 1 m × 300 m) was built. To simulate the weight of rock stratum above the coal seam, 10.94 MPa was exerted on the top boundary. The left and right boundaries of model were set at an equivalent confining pressure of 5.47 MPa. The mechanical parameters of rock and coal stratum, including bulk modulus and shear modulus, are shown in Table 3.

**Table 3.** Mechanical parameters of rock and coal strata.

| Lithology | Rock Stratum | | | | | | Cleats | | | | |
|---|---|---|---|---|---|---|---|---|---|---|---|
| | Density kN/m³ | Bulk Modulus/GPa | Shear Modulus/GPa | Internal Friction Angle/° | Cohesion/MPa | Tensile Strength /MPa | Bulk Modulus /GPa | Shear Modulus/GPa | Internal Friction Angle/° | Cohesion/kPa | Tensile Strength /kPa |
| Coarse grained sandstone | 2410 | 14.44 | 12.22 | 32.00 | 11.80 | 6.03 | 4.55 | 3.58 | 32.00 | 0.01 | 0.02 |
| Fine grained sandstone | 2640 | 18.60 | 18.27 | 28.00 | 21.13 | 10.15 | 4.84 | 3.11 | 28.00 | 0.23 | 0.20 |
| Mudstone | 2420 | 8.33 | 5.74 | 22.00 | 8.87 | 4.32 | 2.06 | 2.06 | 22.00 | 0.10 | 0.47 |
| Aluminous mudstone | 2420 | 8.33 | 5.74 | 22.00 | 8.87 | 4.32 | 2.06 | 2.06 | 22.00 | 0.10 | 0.47 |
| Sandy mudstone | 2220 | 13.68 | 12.03 | 26.00 | 14.76 | 7.52 | 17.78 | 13.68 | 26.00 | 0.45 | 0.41 |
| 4# Coal | 1350 | 0.48 | 0.23 | 23.00 | 4.65 | 3.01 | 0.24 | 0.36 | 23.00 | 0.07 | 0.05 |
| Siltstone | 2530 | 8.06 | 7.63 | 31.00 | 16.77 | 7.12 | 16.52 | 16.12 | 31.00 | 0.06 | 0.08 |
| Conglomerate | 2630 | 16.52 | 15.25 | 30.00 | 16.47 | 8.09 | 4.70 | 3.34 | 30.00 | 0.12 | 0.11 |

### 4.2. Simulation Results

(1) Fracture evolution

The results of overburden fracture evolution during coal mining are shown in Figure 2. From Figure 2a, it can be seen that the direct roof collapsed at a mining distance of 29 m, which might be different from the result of experiments. However, the fracture development process was similar to that in experiments. Therefore, the numerical simulation could also reflect overburden movement. When mining continued to advance to 41 m, roof fractures developed and expanded, causing the basic roof of the rock stratum to break and collapse with the initial weighting. As shown in Figure 2b, fracture development layer continued to increase, and the overburden separation space enlarged and further collapsed in a greater range. At a mining distance of 89 m, the overlying rock stratum collapsed again at a larger scale (Figure 2c). With the working face continuing to move forward gradually, it was rare to see widespread collapse, such as at mining distances of 101 m and 165 m (Figure 2d,e). Before the mining distance of 201 m (Figure 2f), the collapse step was maintained at 12~16 m, which was smaller than the result of experiments.

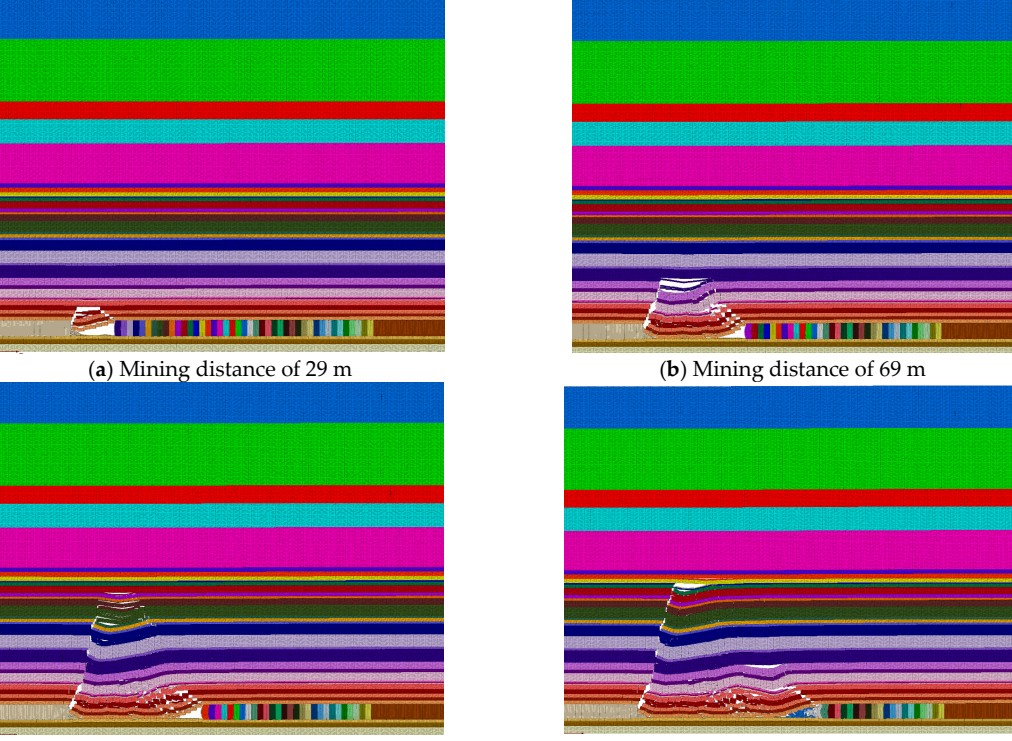

(**a**) Mining distance of 29 m      (**b**) Mining distance of 69 m

(**c**) Mining distance of 89 m      (**d**) Mining distance of 101 m

**Figure 2.** *Cont.*

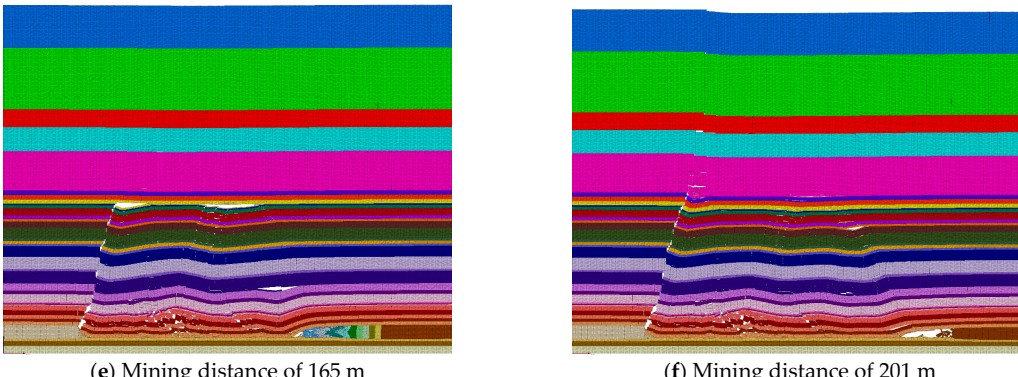

(**e**) Mining distance of 165 m  (**f**) Mining distance of 201 m

**Figure 2.** Fracture evolution of overburden strata in a numerical simulation.

(2)  Stress distribution

The fundamental reason for the evolution of overlying rock fractures was stress accumulation, so stress distribution was analyzed during coal mining, as shown in Figure 3. In the mining process of the 40,205 working face, the stress balance of the overburden influenced by the working face was destroyed and redistributed in real time. After the cut-off of the working face was formed, stress occurred on the front and rear coal wall, as shown in Figure 3a. With the advance of the working face, the stress in the goaf gradually decreased. Before the initial weighting, the roof was not broken, so the overlying strata did exert pressure on the floor. The first weighting occurred when the working face was advanced to 41 m, and the overlying strata began to collapse. Then, the goaf was gradually compacted, and the floor stress in the middle of goaf began to increase, as shown in Figure 3b. Comparing the stress results with fracture evolution, stress increase led to fracture development, which decreased stress in its turn. Therefore, stress in the overburden presented alternating cycles of pressurization, relief, and recompression with mining distance (Figure 3c–f). Stress in the goaf behind the working face tended to be stable.

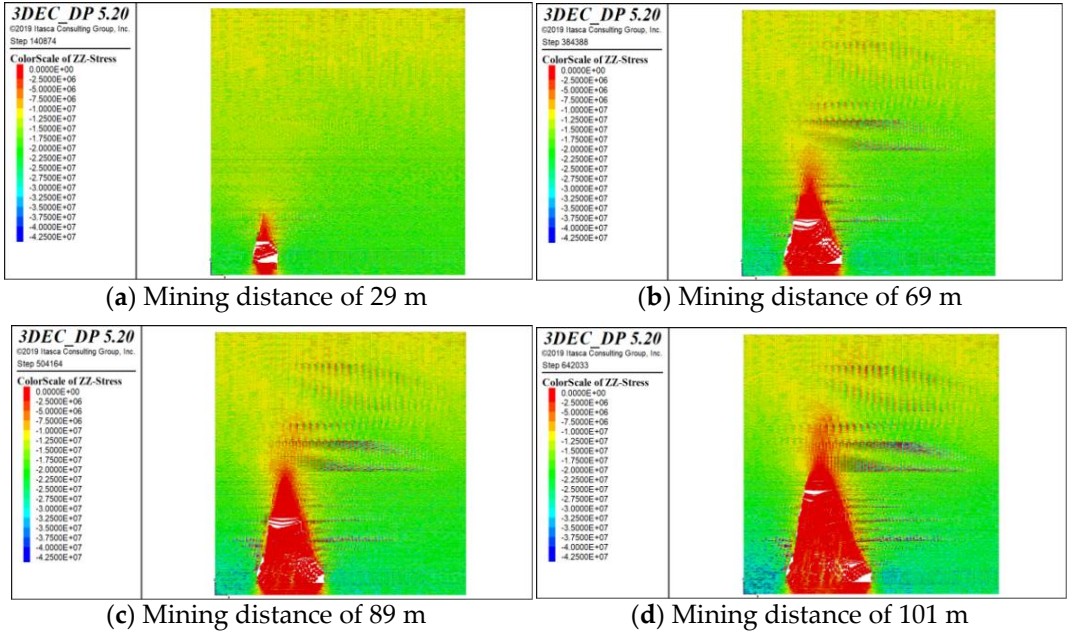

(**a**) Mining distance of 29 m  (**b**) Mining distance of 69 m

(**c**) Mining distance of 89 m  (**d**) Mining distance of 101 m

**Figure 3.** *Cont.*

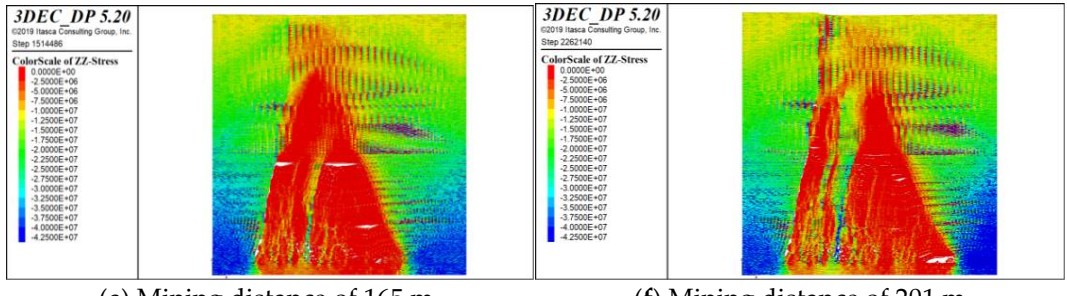

(**e**) Mining distance of 165 m        (**f**) Mining distance of 201 m

**Figure 3.** Stress distribution of overburden strata in numerical simulation.

## 5. Microseismic Response Characteristics

To improve the accurate location of rock fracture development in overlying layers during coal seam mining, microseismic monitoring was adopted. Multiple methods were used to characterize jointly the development of coal seam mining fracture channels.

### 5.1. Layout of on-Site Microseismic Sensors

The mine had a KJ551 microseismic monitoring system, and four seismic geophones were laid in the 40,205 working face. The arrangement of the seismic geophones was as uniform as possible in the plane, and their positions were not arranged in one straight line. The layout position moved forward during face mining.

During the installation of seismic geophones, a 2.8 m anchor bolt (one end of the anchor bolt was processed into M20 screw) was used to drill holes in the vertical roadway roof and the sensors fixed with a resin anchoring agent.

### 5.2. Microseismic Events along the Strike of the Working Face

Abnormal stress leads to fracture evolution, and the process of fracture formation causes microseismic events. What is more, microseismic events within different energies indicated cracks of different scale. At the mine site, microseismic measurements in the mining space were made to determine the fracture range. As shown in Figure 4a, the microseismic response was mainly concentrated between 400 m and 550 m in front of the working face. In the vertical direction, microseismic events occurred about 10–20 m above the roof. Along the dip direction, they were distributed about 20–90 m away from the transport roadway. On 3rd July 2021, the microseismic response changed slightly, but the basic response was consistent. As shown in Figure 4b, the microseismic responses were mainly concentrated between 400–500 m in front of the working face, 7–21 m above the roof, and 0–100 m away from the transport roadway. On 5 July 2021, the microseismic responses were mainly concentrated between 250–550 m in front of the working face, 8–20 m above the roof and 0–100 m away from the transport roadway, as shown in Figure 4c. These results show that the position of microseismic events of high energy indicate instability of the overburden structure, which, due to the influence of advanced stress concentration, was vulnerable to damage. If the working face advanced to this point, the overburden would collapse with mining, and was more likely to form a dominant channel for pressure-relief gas migration. This phenomenon was demonstrated by the microseismic responses on 7 July 2021. On this day, the microseismic events with high energy intensity were mainly concentrated 400–550 m in front of the work face, 7–22 m above the roof, and 0–100 m away from the transport roadway.

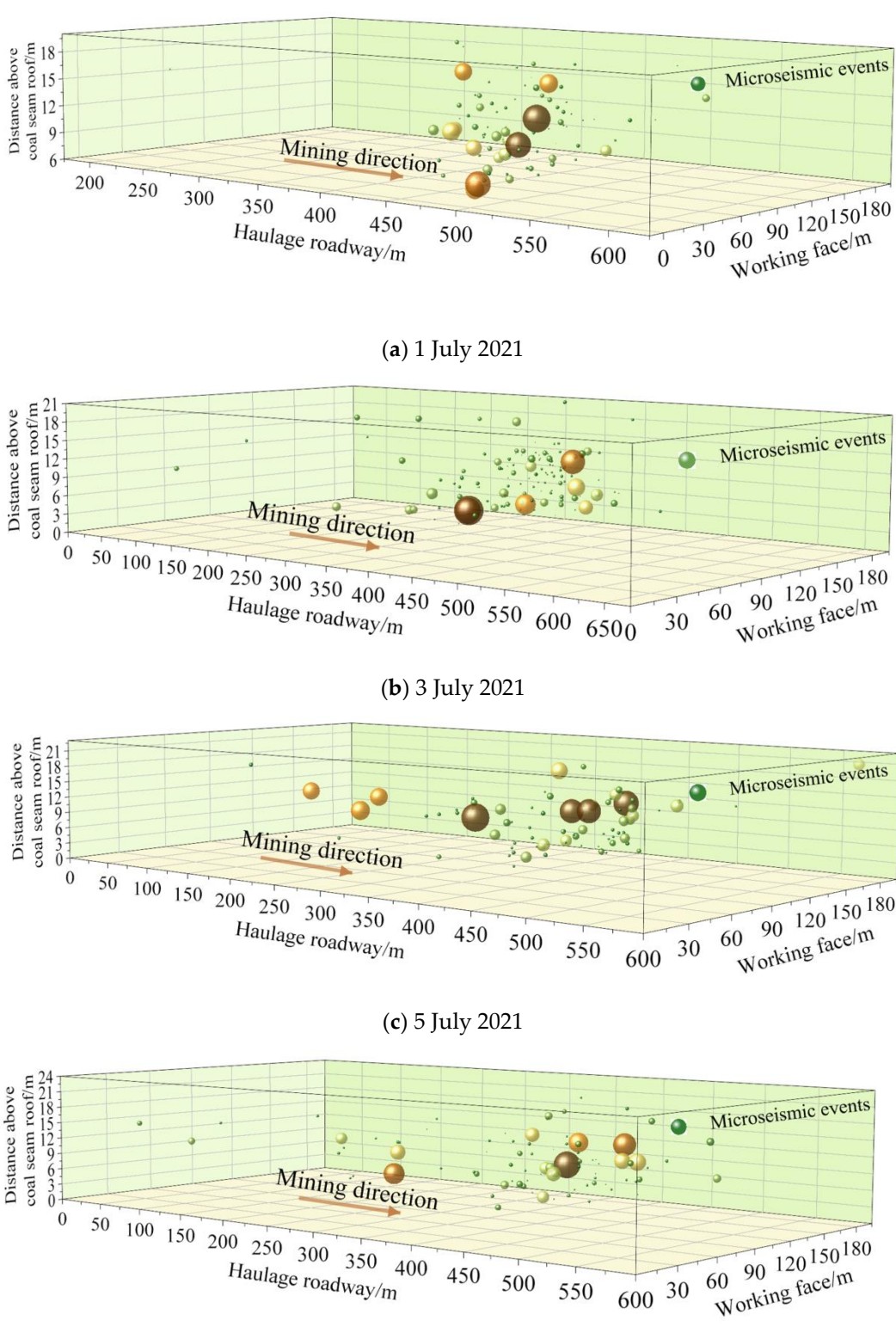

**Figure 4.** Microseismic events along the strike of the working face. (**a**) Microseismic events distribution on 1 July 2021; (**b**) Microseismic events distribution on 3 July 2021; (**c**) Microseismic events distribution on 5 July 2021; (**d**) Microseismic events distribution on 7 July 2021. The size of the ball indicates the magnitude of energy values related to microseismic events.

### 5.3. Microseismic Events along the Dip of Working Face

The distribution of microseismic events in the stope space of the working face can reflect the extent of fracture development. By plotting microseismic events along the dip of working face, the position of fractures can be more accurately ascertained. For example, the microseismic responses were concentrated between 20 and 90 m along the dip of working face, which was consistent with the curve of maximum support working resistance on 1st July 2021, as shown in Figure 5a. It was speculated that due to the mining effects, the support resistance near transportation roadway was than that of the air return lane. The larger stress resulted into more cracks of larger scale, which led to intensive microseismic responses. Figure 5b shows that the maximum support resistance reached a peak 80–90 m away from the transportation roadway, coinciding with the greater energy of the microseismic response at this point. Therefore, microseismic signals may indicate crack ranges and the position of dominant channels. The microseismic events distribution along the dip of working face and support maximum resistance curve were identical on 5 July 2021, as shown in Figure 5c. What is more, the observations were verified by the phenomena on 7 July 2021, as shown in Figure 5d. Intensive and strong microseismic responses are generated in areas severely affected by mining activities. Based on the microseismic monitoring data, distribution of resistance in the support in the working face can be predicted, and the range of dense roof fractures estimated. if the working face advanced to this location, a dynamic disaster would need to be prevented. For gas extraction, it is profitable to determine drilling level, where gas migration channels ensure gas drainage at high concentration.

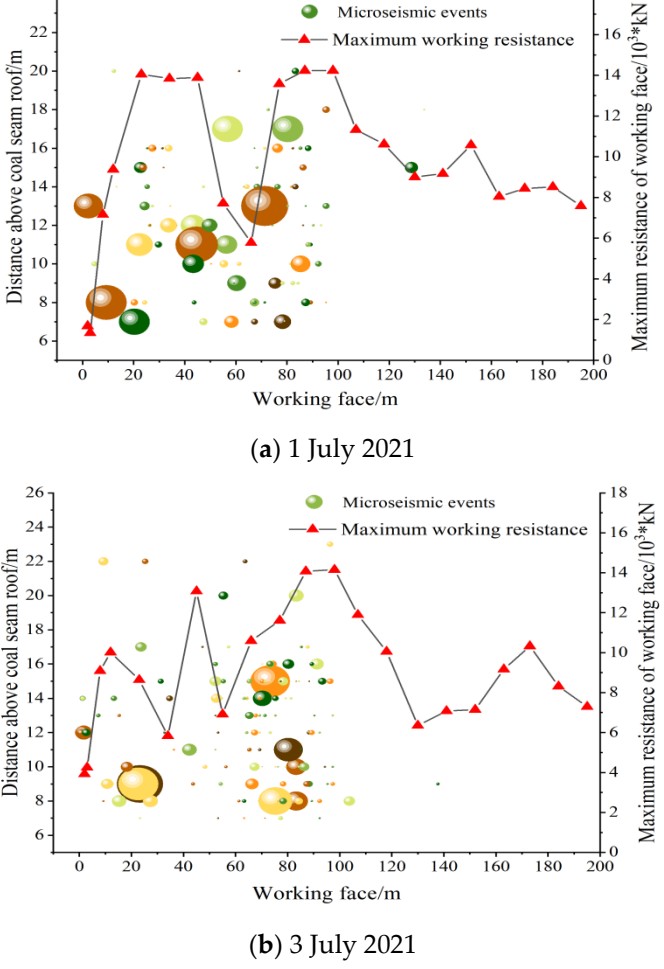

**(a)** 1 July 2021

**(b)** 3 July 2021

**Figure 5.** *Cont.*

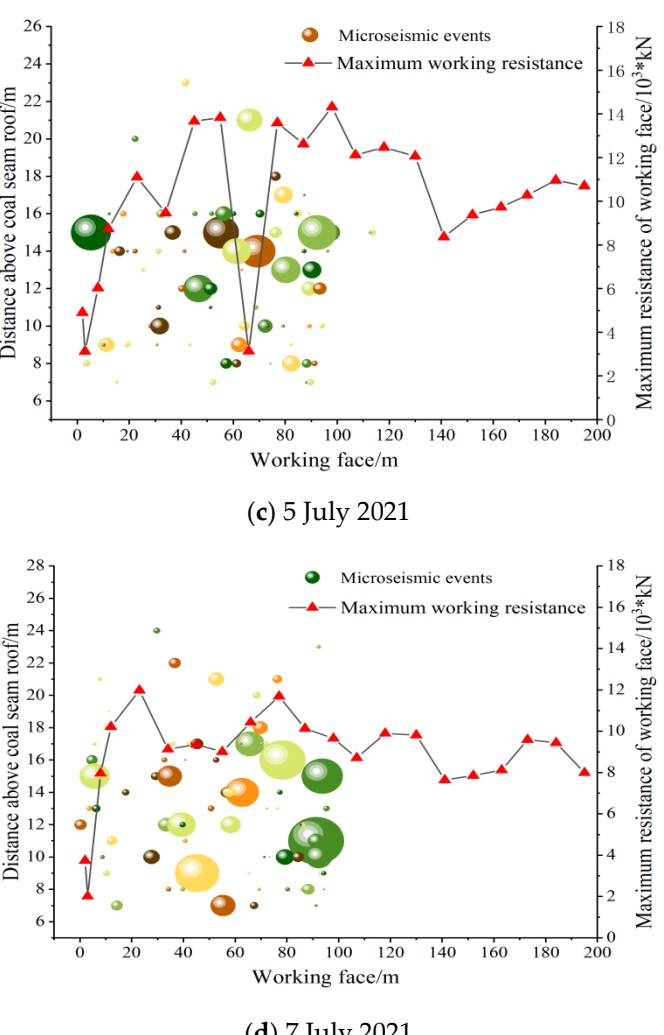

(**c**) 5 July 2021

(**d**) 7 July 2021

**Figure 5.** Microseismic events along the dip of the working face. (**a**) Microseismic events distribution on 1 July 2021; (**b**) Microseismic events distribution on 3 July 2021; (**c**) Microseismic events distribution on 5 July 2021; (**d**) Microseismic events distribution on 7 July 2021. The size of the ball indicates the energy magnitude of microseismic events.

## 6. Discussion

Based on prediction of overburden fracture evolution, gas drainage boreholes can be designed and arranged in advance. With mining distance, the gas migration channels and storage space change with time, so different drainage methods, such as a buried pipeline, a low level borehole, a high level borehole, a high level suction roadway and directional long drilling maybe recommended at different mining stages (Figure 6), as suggested by Lin et al. (2022) [43]. At the initial stage of mining, there are few overburden cracks, and the gas is close to the bottom of the goaf (Figure 1). Therefore, a pipeline method could be used in this stage. With coal mining, fractures form in overburden strata (Figures 1 and 2), and the drainage boreholes need to be arranged in the roof. As overburden strata develop upward, low-level and high-level boreholes may be adopted successively. For thick coal seam mining, the height of the fractures area are higher than in general coal seam mining. What is more, there is a large amount of gas desorbed by mining fracture in coal seams. This requires a drainage method with pumping capacity. A high-level suction roadway can be applied to decrease gas concentration in a short time. Due to longer service cycles and more secure performance, directional long drilling may be used in gas drainage.

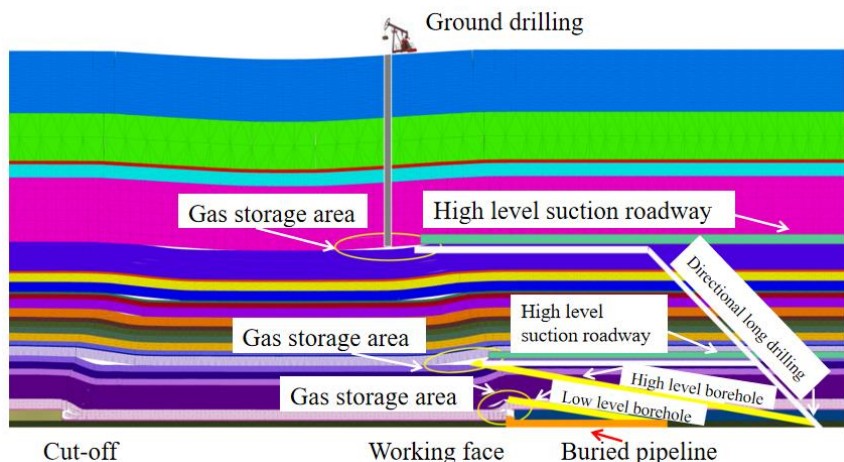

**Figure 6.** Design and layout of extraction boreholes.

## 7. Conclusions

To investigate overburden fracture evolution caused by mining disturbance, experiments and numerical simulations were adopted to characterize the development process, and verified by microseismic monitoring. The main conclusions are as follow:

(1) During coal mining, the roof of the coal seam collapses regularly, and overburden fractures developed in an upward direction. With mining distance increasing, the range and height of the overburden fracture areas enlarge, as observed in numerical simulation. From the numerical simulation, stress distribution indicates the stress increases to exceed the bearing capacity of the rock mass, and the overlying strata breaks and collapses, explaining fracture evolution.

(2) Fracture areas near the working face wall and goaf side provide gas migration channels, while the top fractures area of the overlying strata creates a reservoir space for gas. Therefore, overburden fractures can be used to guide drainage borehole design and construction.

(3) Microseismic responses in the mining space display the spatial distribution of overburden fractures, and verified experimental and numerical simulation observations. Energy and frequency characteristics of the seismic sensors reflect fracture scale and range.

**Author Contributions:** Methodology, Y.C.; formal analysis, Y.C. and X.K.; investigation, Y.C., X.K. and S.L.; resources, S.L. and X.W.; writing—original draft preparation, Y.C. and X.K.; writing—review and editing, P.J. and S.Y.; supervision, X.L., Y.Z. and D.H. All authors have read and agreed to the published version of the manuscript.

**Funding:** This research was funded by National Natural Science Foundation of China (Grant No. 52074217), Natural Science Basic Research Program of Shaanxi (2020JQ-756), China Postdoctoral Science Foundation (2021M693879), and Outstanding Youth Science Fund of Xi'an University of Science and Technology.

**Institutional Review Board Statement:** Not applicable.

**Informed Consent Statement:** Not applicable.

**Data Availability Statement:** The data used to support the findings of this study are available from the corresponding author.

**Conflicts of Interest:** The authors declare no conflict of interest.

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
