# Peer review of "Fracture Evolution of Overburden Strata and Determination of Gas Drainage Area Induced by Mining Disturbance"

_sustainability, doi:10.3390/su15032152_

Round 1

Reviewer 1 Report

Overall Comments

The authors have attempted to study on the important aspect of the evolution of fractures in overburden strata and characterizes it. However, the discussion, validation and analysis is not properly included.

Specific comments

1.     The title itself is confusing, it should be revised to make it more evident on what the readers expect in the paper.

2.     In the abstract, the methodology adopted in not evident, add couple of lines to make it clear.

3.     Line 16: what is physical similar test? not much convincing word?

4.     Line 101-102: ….. It is adjacent to the 40204 goaf at south side, while the north side of it is 40207 working face, which has not been prepared. It is difficult to understand what these numbers indicate?

5.     In Fig. 1, you have shown the gas storage, gas migration etc., which has not been clearly written in the text. Either describe it in the text or in the Figure caption itself.

6.     Fig. 4 and 5: there are no legends in the figure, you have to include them.

7.     There is no discussion section in the manuscript. It seems that the paper merely covers the results obtained from the physical and numerical simulation without validation and analysis of the results.

Author Response

Dear reviewer,

Thank you very much for your suggestions. We have revised them. Please check in the responses and manuscript.

Reviewer 2 Report

o   The authors are trying to target the field issue in the manuscript. However, there must be research background which identify the research problem (paper novelty).

o   The abstract must cover the background, problem/novelty statement, methodology and the prominent results along with the scope of the study.

o   The title and abstract need modification.

o   Show through photos the breakage phenomenon in strata.

o   The authors discussed the literature extensively in the introduction section and the final statement is that the phenomenon is not clear. The authors must clearly identify the problem statement.

o   The authors are just discussing generally the field position.  Discuss the mining position with figures along with excavation and mining sequence. Also, show the project location in map.

o   Strata profile with details.

o   Background and detail description for the data in Table 1.

o   Tests procedures for the Table 2 data.

o   Legend for figure 2.

o   Some issues are within the article with respect to text alignment.

o   Reflect the strata in figure 3.

o   How the authors validated the numerical modelling?

o   The authors have just discussed their data without consulting the literature. Discuss your results along with available literature.

o   The authors have mixed the real field observation with numerical modelling results, physical model tests, and micro seismic observations and reading. The authors must discuss the relevant literature with each approach and discuss the results comprehensively.

o   The conclusion must be concise and be based on the study results. Avoid discussion again and again.

o   Identify the study scope (applications and limitation) along with suggestion for the field applications.

Author Response

Dear reviewer,

Thank you very much for your suggestions. We have revised them. Please check them in response and manuscript.

Reviewer 3 Report

Cai et al. provided an interesting study regarding fracture distribution in overburden and coal mines. Substantial work has been performed, and the results are meaningful. Minor revision is recommended at this stage, and detailed comments are listed below.

1.      Kindly enrich the diversity of the literature review part regarding fracture propeteis and measurements from different perspectives. The article below is suggested to add as a starting point.

Investigations of CO2 storage capacity and flow behavior in shale formation. Journal of Petroleum Science and Engineering 2022. Permeability measurement of the fracture-matrix system with 3D embedded discrete fracture model. Petroleum Science 2022.

2.      Kindly add scale bars in the pictures in figure 1 and figure2.

3.      It is recommended to add an additional explanation below figure 2 so the readers will be quickly informed of the information from the figure.

4.      Kindly add more information regarding the geomechanics model in fig.3.

5.      Please improve the technical writing in the revised manuscript.

Author Response

Dear reviewer,

Thank you very much for your suggestions. We hanve revised them. Please check them in response and manuscript.

Round 2

Reviewer 1 Report

Dear authors

Thank you for the revised version. I have few comments:

1. In Figures 4 and 5, the lgend for small and big balls are not distinguishable. If you can not include legend for all, you can clearly write in the figure caption, what those big and small sized balls are indicating.

2. Discussion part is too short. You have to elaborate it primarily to validate your results.

Good luck

Author Response

Dear reviewer 1

Thank you very much for your comments. We have revised them. Please see them in the attachment.

Reviewer 2 Report

.

Author Response

Dear reviewer 2,

Thank you very much for agreeing to accept our article. In addition, we have asked the professional language service to help us improve the language.